# Does Low-Carbon City Pilot Policy Alleviate Urban Haze Pollution? Empirical Evidence from a Quasi-Natural Experiment in China

**DOI:** 10.3390/ijerph182111287

**Published:** 2021-10-27

**Authors:** Jinling Yan, Junfeng Zhao, Xiaodong Yang, Xufeng Su, Hailing Wang, Qiying Ran, Jianliang Shen

**Affiliations:** 1College of Economics and Management, Xinjiang University, Urumqi 830047, China; yjl200517@163.com (J.Y.); yxdlovezt@126.com (X.Y.); suxufeng84@163.com (X.S.); hailingWANG@xju.edu.cn (H.W.); 2Department of Science, Xinjiang Institute of Technology, Aksu 843100, China; 3Institute of Higher Education, Chongqing Technology and Business University, Chongqing 400067, China; 4Department of Business and Economics, Shanghai Business School, Shanghai 200235, China; ranqiying2021@126.com; 5Center for Innovation Management Research of Xinjiang, Urumqi 830047, China; 6Center for Arid Region Rural Development Research, Xinjiang Agricultural University, Urumqi 830047, China

**Keywords:** low-carbon city pilot policy, haze pollution, quasi-natural experiment, DID model, spatial spillover effect

## Abstract

As a comprehensive environmental regulation, the low-carbon city pilot policy (LCCP) may have an impact on haze pollution. The evaluation of the effectiveness of LCCP on haze pollution is greatly significant for air pollution prevention and control. Taking LCCP as the starting point, in this study we constructed DID, PSM-DID, and intermediary effect models to empirically test the impact and mechanism of LCCP on haze pollution, based on the panel data of 271 cities in China from 2005 to 2018. The findings show that (1) LCCP has significantly reduced the urban haze pollution, and the average annual concentration of PM2.5 in pilot cities decreased by 14.29%. (2) LCCP can inhibit haze pollution by promoting technological innovation, upgrading the industrial structure, and reducing energy consumption. Among these impacts, the effect of technological innovation is the strongest, followed by industrial structure, and energy consumption. (3) LCCP has significantly curbed the haze pollution of non-resource dependent cities, Eastern cities, and large cities, but exerted little impact on resource-dependent cities, Central and Western regions, and small and medium-sized cities. (4) LCCP has a spatial spillover effect. It can inhibit the haze pollution of adjacent cities through demonstration and warning effects. This study enriches the relevant research on LCCP and provides empirical support and policy enlightenment for pollution reduction.

## 1. Introduction

As a result of the surge in energy consumption triggered by the rapid development of China’s urbanization and industry in recent decades, the air quality of cities has deteriorated [1]. The haze pollution caused by fine particles has attracted the extensive attention of the nation. The bulletin of China’s ecological environment in 2019 showed that the concentration of PM2.5 in 337 cities in China was 36 μg/m^3^, the same as that during the same period in 2018. Among these, only 157 cities achieved the air quality standard (i.e., the average annual PM2.5 concentration ≤35μg/m^3^). Although the average annual concentration of PM2.5 in below-standard cities is 40 μg/m^3^, and is decreasing by 2.4% year-on-year, below-standard cities still account for 53.4% of all cities (see Figure 1). The 2020 work report of the Chinese government noted that it is imperative to further improve the effectiveness of ecological environment treatment and highlighted the legal, scientific, and accurate pollution control. Accordingly, identifying means to effectively curb the continuous deterioration in haze pollution and improve air quality has become one of the important issues for China’s green economy development and ecological civilization construction.

The frequent occurrence of hazy weather may be induced by the release of a large number of fine particles (i.e., PM2.5) from fossil energy consumption, especially the combustion of coal [2]. The public product characteristics of environmental protection issues determine the main responsibility status of government departments in providing these products. Although the Chinese government has issued a series of anti-haze measures, haze control is not as effective as expected by citizens, which also reflects the lack of efficiency and the restraints on the government’s haze control. Therefore, due to the public characteristics of environmental problems, haze pollution should be alleviated through government regulation. As a green, compact, and intensive urban development model, low-carbon urban development may reduce haze pollution and improve environmental pollution by cultivating low-carbon industries, developing a circular economy, using clean energy, and promoting sustainable transportation [3,4]. Globally, countries have adopted the strategy of LCCP as the core to coping with environmental pollution. China is an important contributor to climate change strategy, and the National Development and Reform Commission (NDRC) launched three stages of LCCP projects in 2010, 2012, and 2017. In 2010, the NDRC launched the first LCCP project in five provinces and eight cities. In 2012, in order to further carry out the target of “building a beautiful China”, the NDRC launched the second project in 29 provinces and cities, expanding the scope of LCCP projects. According to the positive results of the first two pilot projects, the NDRC implemented the third stage in 2017 to further implement the 13th Five-Year plan. At present, the low-carbon pilot cities are spread across China (see Figure 2). It can be seen that central and local governments have paid increasing attention to the development of low-carbon cities. Accordingly, some of the core issues of this study are as follows: Can LCCP reduce urban haze pollution? How does LCCP reduce haze pollution? Is there heterogeneity in the effect of LCCP on haze pollution? Discussions regarding these questions have important practical significance for improving environmental quality and promoting the construction of ecological civilization.

The possible contributions of this paper encompass four aspects: (1) LCCP and haze pollution are included in the same research framework, which provides new research ideas for reducing haze pollution. (2) From the multiple perspectives of technological innovation, industrial structure, and energy consumption, this paper theoretically explains the internal transmission mechanism and empirically tests the specific channels of the effect of LCCP on haze pollution, so as to provide a path choice for further optimizing LCCP. (3) The multi-dimensional heterogeneity analysis of urban resource dependence, regional characteristics, and scale characteristics are undertaken to determine the differential impact of LCCP on haze pollution. (4) Considering the spatial attribute of haze pollution, in this study we adopted spatial econometric methods to further investigate whether LCCP has a spatial spillover effect on haze pollution in neighboring areas.

Following this introduction section, Section 2 provides a literature review. The theoretical analysis and hypotheses follow in Section 3. Section 4 presents the data and model framework. Section 5 presents the analysis and discussion of the empirical results. In Section 6, further analysis of the spatial effect of LCCP on haze pollution is undertaken. Section 7 provides the conclusions and implications.

## 2. Literature Review

Many studies have found that air pollution not only affects the health of residents, but also may reduce residents’ subjective well-being, damage cognitive function, and produce negative emotions and behaviors [5,6]. The exacerbation of urban haze pollution and its serious consequences have stimulated scholars’ enthusiasm for the research on the causes [7], formation mechanisms [8], sources [9,10,11], and influencing factors of haze pollution [12,13,14,15]. Generally, the influencing factors include economic growth [16,17,18], transportation construction [19,20,21], urbanization [22,23], industrial agglomeration [24,25], energy intensity [5,16,26,27], FDI [28], trade liberalization [29], technological progress [30], environmental regulation [31,32,33], and other meteorological factors [34,35,36]. Xu and Lin [5] adopted panel data models to study the main driving forces of PM2.5, involving economic growth, urbanization, private cars, coal consumption, and energy efficiency, and also analyzed the regional heterogeneity in China. Chen et al. [16] took countries with different income levels as the research target to study the causal relationship between energy consumption, energy intensity, economic growth, urbanization, and PM2.5 concentration. Zhou et al. [19] utilized a geographically weighted regression (GWR) method to analyze how haze pollution is affected by factors including technical inputs, industrial output values, and public transportation. Cheng et al. [28] adopted a dynamic spatial panel model to empirically study the impact of FDI on PM2.5. Yi et al. [30] constructed a theoretical framework to study the impact of heterogeneous technological progress on haze pollution. Zhang et al. [32] used the GMM (Generalized Method of Moments) method to analyze the different types of environmental regulations on haze pollution. Xu et al. [5] utilized a Weather Research and Forecasting/Community Multiscale Air Quality (WRF/CMAQ) modeling system to study the impact of meteorological conditions on PM2.5. In summary, existing studies have fully examined haze pollution and its influencing factors. However, the deterioration in haze pollution requires intervention by governments and implementation of urban construction policy [32].

Urban construction policy is an important factor affecting environmental pollution [13]. Previous literature has studied haze pollution from the perspective of urban construction policy, but few studies have mentioned the impact of low-carbon policy. LCCP is a new and important model for urban healthy development. Its main purpose is to reduce energy consumption, lower pollution emissions, protect the urban ecological environment, and realize green and sustainable development [37]. A series of academic evaluations of the implementation effect of LCCP suggests that the impacts of LCCP on economic growth and environmental pollution are becoming popular research topics. Song et al. [37] evaluated three batches of low-carbon pilot cities in China by constructing an evaluation index system of low-carbon city construction based on six dimensions of energy, industry, low-carbon life, etc., finding that LCCP plays a positive role in facilitating urban low-carbon development, but there are local imbalances in the evolution of low-carbon cities. Feng et al. [38] adopted the difference-in-differences (DID) approach to study the effects of LCCP, finding that it can significantly increase carbon intensity. Fu et al. [39] found that LCCP improves the overall carbon emission efficiency of pilot cities, and is more effective in Eastern pilot cities. Using the difference-in-differences (DID) model, Lin et al. [40] empirically found that LCCP greatly reduced API and other air pollution indexes and improved urban air quality. Cheng et al. [41] found that China’s LCCP has a positive effect on green economic growth and has a more obvious impact on Eastern cities. Chen et al. [42] utilized PSM-DID and other methods to test whether and how the low-carbon city pilot policy affects enterprise TFP, and found that low-carbon cities significantly promote an increase in the TFP of local enterprises. In addition to the economic and environmental effects resulting from the pilot policy, other benefits also provide important perspectives to evaluate the effectiveness of LCCP. Ma et al. [43] empirically found that LCCP has improved the level of green technology innovation overall, based on Green Patent Data of Chinese A-Share Listed Companies. These studies are conducive to our understanding of China’s LCCP.

In summary, the existing research has provided good ideas for this study. As a comprehensive environmental policy tool, LCCP may have an impact on haze pollution. However, previous studies have paid less attention to this aspect. Accordingly, in this study, we took LCCP as the starting point, and constructed DID, PSM-DID, and intermediary effect models to empirically test the impact and mechanism of LCCP on haze pollution. This was based on the panel data of 271 cities in China from 2005 to 2018. The study enriches the relevant research on LCCP and haze pollution, providing empirical support and policy enlightenment for pollution reduction.

## 3. Theoretical Analysis and Research Hypothesis

LCCP is an important part of China’s national environmental governance strategy system [41]. It aims to achieve low-carbon development in the context of high pollution and high energy consumption by reducing the proportion of high-carbon industries. Theoretically, it can be used to fully utilize the synergy effect between climate change and energy conservation, environmental protection, new energy development, and ecological construction; actively explore the institutional mechanisms conducive to energy conservation, pollution reduction, and the development of low-carbon industries; implement the target responsibility system for controlling pollutant emissions; and study how to use market mechanisms to achieve pollution reduction targets. In addition, LCCP can accelerate the establishment of an industrial system featuring low carbon emissions. Pilot cities can accelerate low-carbon technology innovation, promote low-carbon technology R&D, actively use low-carbon technology to transform and upgrade traditional industries, and cultivate and expand strategic emerging industries such as new energy. Furthermore, LCCP can closely track the most recent progress in low-carbon technology, and actively promote technology introduction, digestion and absorption, re-innovation, and joint research and development with foreign countries. LCCP can also be used to establish the statistics and management system relating to pollution emission data, and a complete data collection and accounting system to strengthen the supervision and control of pollution discharge, and thus provide an institutional guarantee for pollution treatment.

Accordingly, the following hypothesis is proposed:

**Hypothesis (H1).** 
*LCCP can effectively reduce haze pollution.*


The above analysis shows that LCCP may effectively reduce haze pollution. If the effect of haze reduction does exist, what is the mechanism by which haze pollution is reduced?

According to the research of Grossman and Krueger [44] and Brock and Taylor [45], the main means by which the environmental quality are affected include the scale effect, the structure effect, and the technology effect. Therefore, this study assumed that LCCP may reduce haze pollution by promoting technological innovation (innovation effect), optimizing industrial structure (structure effect), and reducing energy consumption (scale effect). The mechanism diagram of the impact of LCCP on haze pollution is shown in Figure 3.

(1) LCCP can reduce haze pollution by promoting technological innovation. The “innovation compensation” theory states that rigid environmental supervision policies enable enterprises to raise technological R&D investment and change production methods to compensate for the economic loss caused by environmental protection costs [37]. The essence of implementing environmental policy is to exert innovation pressure on enterprises and force them to make independent innovation, so as to improve the productivity and market competitiveness of enterprises. LCCP increases the pollution control and production costs of enterprises. As enterprises seek maximum benefits, the added cost will incentivize enterprises to enhance their creativity. Accordingly, LCCP offers a “continuous incentive” for the technological innovation of enterprises [42]. In addition, however, a major target of LCCP is to build a low-carbon scientific and technological innovation mechanism, promote production, energy conservation and environmental protection technologies through technological innovation, and apply new technologies to pollution discharge and control of enterprises. Therefore, as a tool of environmental regulation policy, LCCP is consistent with Porter’s “innovation compensation” theory.

**Hypothesis (H2).** 
*LCCP can reduce haze pollution by promoting technological innovation.*


(2) LCCP can inhibit haze pollution by optimizing the industrial structure. The structure effect of LCCP is primarily manifested in forcing the transformation and upgrading of industrial structure [37] and improving environmental quality. Traditional industries depend on the massive investment of capital and labor factors, and LCCP will strengthen the upgrade and adjustment of these industries to those with high technology, high added value, low pollution, and low energy consumption, and eliminate enterprises with high energy consumption and high pollution, hence providing greater development space for enterprises with low energy consumption and low pollution. Under the guidance of LCCP, the traditional pollution-intensive industries will be phased out and low-carbon industries consistent with the city’s advantages will be developed; examples are low-carbon agriculture, the new energy industry, and emerging industries represented by green and low-carbon technologies [46], which clearly promote the optimization and upgrading of the industrial structure. Therefore, LCCP not only has an effect on the industrial structure, but also has the task of industrial structure optimization, which is conducive to reducing haze pollution.

**Hypothesis (H3).** 
*LCCP can reduce haze pollution by upgrading industrial structure.*


(3) LCCP can reduce haze pollution by encouraging market players to reduce energy consumption in various ways. LCCP is an environmental regulation tool aimed at reducing the environmental pollution. Under the constraints of LCCP, local governments, production enterprises, and the public will reduce energy consumption. Specifically, local governments will adopt a scientific and efficient management model for urban consumption, transportation, and living systems to improve energy efficiency [47]. Facing the high cost of environmental governance, enterprises will “force” themselves to recognize their technological shortcomings, improve production and environmental protection technologies, and adjust the energy consumption structure, so as to reduce energy consumption. In addition, LCCP can urge individuals and families to practice green and low-carbon lifestyles, actively use low-carbon products, and choose low-carbon travel modes by vigorously publicizing the green and low-carbon concepts and developing low-carbon public transportation systems [48]. These not only help to reduce energy consumption intensity but also further improve environmental quality, eventually reducing haze pollution.

**Hypothesis (H4).** 
*LCCP can reduce haze pollution by decreasing energy consumption.*


## 4. The Model Setting, Variables and Data Description

### 4.1. Model Setting

LCCP was implemented as a quasi-natural experiment, because the first batch of pilot cities was mostly selected from provincial administrative regions, and the third batch was launched later. Therefore, based on the second batch, which was selected in 2012, we construct two dummy variables, taking pilot cities as the treated group, and non-pilot cities as the control group. The model is constructed as follows:(1)$$\ln PM2.5=\beta_{0}+\beta_{1}LCCP_{it}+\beta_{2}X_{i}{}_{t}+\mu_{i}+\lambda_{t}+\varepsilon_{it}$$
where *i* is the city and *t* is the year; lnPM2.5 indicates the regional haze pollution level; *LCCP* is a dummy variable, indicating whether the low-carbon pilot policy is implemented; *X* represents a series of control variables; $\beta_{1}$ represents the net effect of LCCP on haze pollution,$\mu_{i}$ and $\lambda_{t}$ represent individual fixed effect and time fixed effect, respectively; and $\varepsilon_{it}$ represents random interference terms.

LCCP may reduce the level of urban haze pollution. According to the above theoretical mechanism analysis, we assume that LCCP can reduce haze pollution through technological innovation, industrial structure, and energy consumption. In order to verify these assumptions, in this study we used the three-step verification method proposed by Baron and Kenny [49], and the specific model is set as follows:(2)$$\ln PM2.5_{it}=c_{1}+\theta_{1}LCCP_{it}+\theta_{2}X+\mu_{i}+\lambda_{t}+\varepsilon_{it}$$
(3)$$M_{it}=c_{2}+\eta_{1}LCCP_{it}+\eta_{2}X+\mu_{i}+\lambda_{t}+\varepsilon_{it}$$
(4)$$\ln PM2.5_{it}=c_{3}+\varphi_{1}LCCP_{it}+\varphi_{2}M_{\mathrm{it}}+\varphi_{3}X+\mu_{i}+\lambda_{t}+\varepsilon_{it}$$

### 4.2. Variable Selection

#### 4.2.1. Explained Variable

Haze pollution (PM2.5)—In contrast to the use of “three industrial wastes” and other indicators to measure haze pollution, as used in most literature, and considering that the main “culprit” of haze pollution is PM2.5, in this study we used the concentration of fine particulate matter PM2.5 to measure haze pollution, so as to more accurately and objectively measure the environmental situation of a region, particularly the atmospheric environmental pollution. In this study, according to the annual World PM2.5 density data from 2005 to 2018 shared by the atmospheric composition analysis group (ACAG) of Dalhousie University, after grid processing and mask processing in line with the map of China in ArcGIS, the annual average data of PM2.5 concentration in prefecture-level cities in China was calculated.

#### 4.2.2. Core Explanatory Variable

Low carbon city pilot policy (LCCP)—LCCP is defined as the value of 1 in the year in which a city implements LCCP and each year after that; otherwise, it is 0. This definition automatically produces the treated group and the control groups, in addition to the double differences before and after the implementation of the policy, which is equivalent to the cross term of the treated city variable

#### 4.2.3. Control Variables

(1) Economic development level (RGDP) is measured by per capita GDP [17]. RGDP can reflect the economic development of a region, and the economic development will inevitably affect the air pollution of the region. (2) Human capital (HUM) is measured by the ratio of the number of students in colleges and universities to the total population at the end of the year. A higher human capital level is conducive to the absorption, promotion, and application of new technologies, and thus to the improvement in environmental quality [13]. Additionally, cities with high human capital have a higher expectation of better environmental quality and will force local governments to carry out strict environmental supervision measures to reduce pollution discharge. (3) Foreign direct investment (FDI) is expressed by the amount of foreign capital used [28]. FDI can not only increase the local capital stock but also promote the application and management of local advanced technology through technology linkage and knowledge spillover, thus reducing pollutant emissions. However, some developing countries attract foreign investment by lowering the threshold of environmental regulations in order to pursue short-term economic benefits, making their regions a “pollution haven” for developed countries. (4) Government scale (FIN) is measured by the proportion of government expenditure in the national economy [33]. Local governments are both facilitators and regulators of environmental pollution. The scale and structure of local government financial expenditure have a great influence on reducing the environmental pollution. (5) Information level (INFOR) is measured by the proportion of Internet users in the urban population. In cities with a high degree of information technology, the government and the local population have a better grasp of local air pollution [36]. Although these factors enhance the awareness of environmental protection, law enforcement departments can also easily find the pollution sources to reduce air pollution.

#### 4.2.4. Mediating Variables

(1) Technological innovation (PATENT)—Technological innovation aims to improve the utilization of resources, and achieve energy conservation and emission reduction, which has reached a consensus in academic circles. It is represented by the number of patent applications authorized.

(2) Industrial structure (IND)—Industrial structure connects production activities with the environment, and determines the energy consumption mode and pollutant emission level in the economic system, and thus has a substantial impact on the environment. It is denoted by the proportion of the output value of the tertiary and secondary industries.

(3) Energy consumption (PELE)—China’s traditional coal-based energy structure determines that energy consumption will emit a large number of pollutants, resulting in the deterioration in urban environmental quality. Therefore, the degree of energy consumption will have a significant impact on environmental pollution. It is measured by the per capita power consumption of each city. Due to the lack of data on energy consumption at the urban level, and considering the high correlation between power consumption and energy consumption, in this study we measured energy consumption by the per capita power consumption [50].

### 4.3. Data Description

Restricted by the availability of data and consistency of the statistical indicators, this study selected 271 cities in China from 2005 to 2018. Among these, data relating to the low-carbon pilot cities were taken from the data released by the NDRC. Other data were from the National Statistical Yearbook and the statistical yearbooks of provinces and cities for various years. In addition, in order to control the heteroscedasticity of the model, all variables were processed by a natural logarithm. The descriptive statistics of relevant data are shown in Table 1 below.

## 5. Analysis of Empirical Results

### 5.1. Parallel Trend Test

In order to ensure the effectiveness of the regression results, the DID model requires that the treated and control groups have the same change trend before the implementation of the policy [51]. In view of this, this section first decomposes and analyzes the dynamic trend. Firstly, taking the implementation of the LCCP as the base year, the variable Ad is set in the year affected by the policy as AD0 = 1, otherwise it is 0, and the abscissa in the figure is 0. In the second year after being affected by the policy, AD1 = 1, otherwise it is 0; in the figure, the abscissa is 1, and so on. Similarly, in the first year affected by the policy, AD-1 = 1, otherwise it is 0, and the abscissa in the figure is −1. In the second year before being affected by the policy, AD-2 = 1, otherwise it is 0, and so on. Then, the parallel trend and dynamic effect test were carried out, and Figure 4 was drawn. The abscissa in Figure 4 is the continuous year of the sample city affected by the policy, abscissa 0 represents the year of policy implementation, and “negative” represents the year before policy implementation. The results show that before the implementation of LCCP, the estimated coefficient fluctuates around the ordinate of −0.5, which shows that the difference between the treated group and the control group was not obvious before the implementation of the policy, and thus meeting the condition of equilibrium trend. However, the “policy effect” after the implementation of the policy is very obvious. This conclusion confirms that the DID model satisfies the hypothesis of a parallel trend.

### 5.2. Analysis of Benchmark Regression Results

Firstly, the Hausman test was conducted. The result shows that the *p*-value is less than 0.05, rejecting the original hypothesis, so the fixed effect model is suitable for regression. In Table 2, columns (1)–(6) show the regression results of incrementally adding control variables. The results show that regardless of how the control variables are added, LCCP has a significant negative impact on PM2.5, suggesting that LCCP significantly reduces urban haze pollution, and H1 is confirmed. From the regression results of control variables, the level of economic development significantly reduced haze pollution, indicating that, in the cities with a high level of economic development, people pay more attention to personal health and the awareness of environmental protection is stronger, which is conducive to haze pollution reduction. The coefficient of FDI is significantly positive, which verifies the existence of a “pollution paradise” in China [52]. Since the reform and opening up, under the dual pressure of environmental protection and official promotion, the Chinese government has been more likely to undertake some foreign pollution-intensive industries to develop the local economy. This will inevitably lead to more serious haze pollution in cities with greater foreign direct investment, making the haze reduction effect of LCCP less significant. The government’s fiscal expenditure coefficient is significantly negative, indicating that the government will subsidize enterprises and encourage them to invest more funds in R&D of new technologies, especially polluting gas clean treatment technology, which helps to reduce haze pollution. The information level of the city significantly reduces haze pollution. The reason for this is that in cities with a high informatization degree, the local government can more comprehensively determine the haze pollution through the information network, the government has a better grasp of the local air pollution situation, and the law enforcement department can easily find pollution sources and strengthen the haze pollution control, avoiding hidden economic and regulatory loopholes.

### 5.3. Estimation Results of PSM-DID Method

In order to reduce the estimation error, the DID method was used to address the systematic differences in the changing trend between pilot cities and non-pilot cities [53]. Through the PSM method, the difference in the haze pollution level caused by systematic differences was reduced to the greatest extent, so as to reduce the error of DID estimation. Before using the PSM-DID method, a test should also be undertaken to determine if PSM method is effective. First, we should see whether the hypothesis of common support is tenable. As can be seen from Table 3, no significant differences were found in all control variables after being matched. Secondly, Figure 5 shows that the probability density distributions of the treated and control groups are relatively close, indicating a good matching effect. Thus, it is feasible and reasonable to employ the PSM-DID method.

In order to solve endogenous problems, the PSM-DID method was again used for regression. The results are shown in Table 4. LCCP still significantly reduces the haze pollution level. The coefficient and direction of the regression results are roughly consistent with the DID regression results, further verifying the robustness of the regression results.

### 5.4. Mechanism Test of LCCP on Haze Pollution

The above empirical results indicate LCCP can significantly reduce the level of urban haze pollution, but how does LCCP reduce haze pollution? The three-step verification method in Equations (2)–(4) was conducted to test the influencing mechanism of LCCP on haze pollution. Table 5 displays the results. In column (1), the haze reduction effect of LCCP is −0.1429 and significant at the level of 1%, consistent with the conclusions in the benchmark regression. In column (2), the regression coefficient of LCCP on technological innovation is significantly positive at the level of 1%, indicating that LCCP is conducive to urban technological innovation. In column (3), the regression coefficient of technological innovation on haze pollution is significantly negative at the level of 1%, indicating that the level of technological innovation can effectively reduce haze pollution. Combined with the above results, LCCP improves the level of urban technological innovation, and technological innovation inhibits haze pollution, of which the intermediary effect is 0.0205, accounting for about 14.36% of the total effect. The Sobel test and bootstrap test are significant at least at the level of 5%, indicating that there is clearly an intermediary role of technological innovation. Columns (4)–(5) and columns (6)–(7) are the regression results when industrial structure and energy consumption, respectively, are taken as intermediaries. The intermediary effects of industrial structure and energy consumption are −0.0189 and −0.1159, respectively, accounting for about 13.23% and 4.06% of the total effect. These results verify H2, H3, and H4. Furthermore, the effect of technological innovation is the strongest, followed by that of industrial structure and energy consumption. The results further imply that LCCP has a good effect on haze reduction through technological innovation and industrial structure upgrading, whereas the effect of energy consumption is weak, and there is still much room for improvement.

### 5.5. Robustness Test

In order to improve the robustness of this research conclusion, in this study we carried out a robustness test considering the following four aspects: ① The placebo test—The hypothesis is that if the pilot city is not affected by LCCP, the changing trend of pollution in the control and treated groups will not have a systematic difference over time. The regression is then conducted again based on this hypothesis. The results show that the estimation coefficient of the LCCP is negative, but not significant, indicating that the above benchmark regression is caused by the exogenous impact of LCCP, rather than the placebo effect caused by other factors. ② Replace the explained variable and re-regress with SO2 as the explained variable, and the coefficient and significance of the regression results were roughly consistent with the benchmark regression. ③ After excluding 31 key cities, such as provincial capital cities and sub-provincial cities, the results were still basically consistent with the previous conclusions. ④ All variables lagged for one period. Considering that LCCP may not have an immediate impact, in addition, in order to avoid the error of simultaneous equations, the lags for one period of LCCP and all control variables are substituted into the equation for regression. The regression results are roughly consistent with the previous text, verifying that the conclusions of this paper are robust. See Table 6 for specific results.

### 5.6. Heterogeneity Test

#### 5.6.1. Heterogeneity Analysis of Urban Resource Dependence

Due to the significant differences in natural resource endowment, economic development, and industrial structure in Chinese cities, the impacts of LCCP on haze pollution may vary. In order to deeply investigate the heterogeneity of urban resource dependence, according to the national sustainable development plan of resource-based cities (2013–2020), prefecture-level cities in China were divided into resource-based cities and non-resource-based cities. The regression results are shown in Table 7. Columns (1)–(2) show the results of non-resource-based city samples, and (3)–(4) show the results based on resource-based city samples. It can be observed that for the non-resource-based cities, the impact of LCCP on haze pollution is significantly negative at the level of 1%, indicating that LCCP inhibits haze pollution. However, for resource-based cities, the coefficient of low-carbon pilot policy is not significant.

#### 5.6.2. Analysis of Urban Regional Heterogeneity

Due to China’s unique geographical conditions and differences in resource endowment, the haze reduction effects of LCCP on China’s Eastern, Central, and Western regions may also be quite different. Because the number of cities with LCCP in the western region is relatively few, in this study we divided the research samples into two parts: the eastern regions and the central and western regions. The regression results are shown in Table 8: the regression coefficient in the East is significantly negative at the level of 1%, whereas the coefficient in Central and Western regions is not significant.

#### 5.6.3. Analysis Results of Urban Scale Heterogeneity

In addition, the size of cities plays an important role in the impact of LCCP on reducing haze pollution, which may also lead to differences. Based on the data of permanent residents in urban areas of local cities in 2015, cities were divided into three sizes: large-sized, medium-sized, and small-sized cities according to the notice on adjusting the criteria for urban scale division issued by the State Council (GF [2014] No. 51). As shown in Table 9, the LCCP coefficient of large cities is significantly negative, whereas those of small- and medium-sized cities are not significant, indicating that LCCP in large cities has better haze reduction effects and these effects decline with the decrease in city size.

### 5.7. Discussion on the Empirical Results

#### 5.7.1. Discussion on the Benchmark Regression Results

LCCP significantly reduces urban haze pollution and the average annual concentration of PM2.5 in pilot cities decreased by 14.29%. LCCP is aiming to promote the construction of an ecological civilization, and green and low-carbon development [39]. Its core theme is to emphasize the long-term coordinated development of ecological environment protection and economic growth. The Chinese government has made provisions in the specific tasks of the pilot cities, such as establishing a target responsibility system for controlling pollutant emissions, optimizing the industrial structure, establishing a low-carbon industrial system, and strengthening talent team building for low-carbon development. Under this guidance, each pilot city has formulated a more operational and regional low-carbon development model, path, and policy system, which are connected with its industrial factor endowment and economic development stage. By promoting technological innovation, optimizing the industrial structure, improving energy efficiency, and reducing the proportion of industries with high energy consumption, high pollution, and high emissions, LCCP effectively takes advantage of the “positive external effects” [42]. In this way, haze pollution is significantly reduced.

#### 5.7.2. Discussion on the Mechanism Test

Table 5 shows that LCCP has a positive effect on haze reduction through technological innovation, upgrading the industrial structure, and decreasing energy consumption.

The technological effect is a significant contributor to the improvement in environmental quality. Its function mainly lies in the good effect of energy conservation and emission reduction resulting from new technological progress [54]. LCCP will tighten urban environmental constraints. This will force enterprises to increase the resources dedicated to the R&D, innovation, and application of new technology in the field of production and environmental protection. In addition, LCCP will lead to technological advance, which contributes to the improvement in pollution protection and treatment. For example, it can enhance the ability to systematically observe, collect, and monitor relevant pollution data, which play an important role in closely identifying the high-polluted industries and important pollution sources, so that pollution prevention and control can be timely and effectively carried out.

Transforming and upgrading the traditional industrial structure is one of the main tasks of LCCP to improve environmental quality. Most traditional industries depend on the massive investment of capital and labor factors. Guided by LCCP, these industries will be upgraded and transformed into industries characterized by advanced technology, high added value, and low pollution. In this manner, the high-polluting enterprises will be forced out of business, and some favorable policies and financial aid will be implemented for low-polluting enterprises. As a result, strategic emerging industries, such as new energy, will be further fostered and expanded, and ultimately a low-polluting and low-emitting green industrial system will be established [55].

LCCP can reduce energy consumption. As an environmental regulation tool, LCCP is designed to reduce pollutant discharge. Due to the higher costs associated with pollution control and treatment, enterprises will be “forced” to reduce energy consumption and improve energy utilization. Specifically, energy consumption will emit a large number of pollutants, resulting in the deterioration of urban environmental quality. LCCP is dedicated to advocating the green and low-carbon concept and developing low-carbon transportation systems. In this manner, the energy structure is optimized and the energy consumption is reduced. Additionally, under the guidance of LCCP, citizens will adopt scientific and efficient forms of consumption, transportation, and lifestyle, thus reducing energy consumption and improving energy efficiency. Third, LCCP establishes a carbon credits system which specializes in the promotion of energy utilization and reduction of pollutant emissions.

#### 5.7.3. Discussion on the Heterogeneity Test

The impact of LCCP on haze pollution may vary due to the significant differences in natural resource endowments, stages of economic development, and industrial structures in Chinese cities.

In terms of resource dependence heterogeneity, the possible reason for this is that cities that are highly dependent on natural resources tend to develop resource-based industries. Due to long-term and high-intensity mining, resource-based cities generally face problems such as environmental pollution and ecological destruction [56]. In addition, in regions with an abundance of natural resources, the allocation of production factors, such as labor and capital, among resource exploitation departments, and technology research and development departments, is severely distorted, so the effect of low-carbon pilot policies on haze reduction is not obvious.

A possible reason for regional heterogeneity is that the Eastern regions have a high-level economic development and relatively complete infrastructure, which provides a material guarantee for the effective implementation of LCCP [57]. In the Central and Western regions, due to the relatively backward economic development, the extensive economic development mode leads to rapid economic development at the expense of the environment. In addition, compared to the Eastern regions, there is a large gap in technology, human capital, and capital allocation in Central and Western regions. Therefore, LCCP is less effective at reducing haze pollution in the Central and Western regions.

Urban scale heterogeneity occurs because, compared with small and medium-sized cities, large-sized cities have greater R&D investment, and better implementation of relevant environmental protection systems and policies, which can enhance haze pollution control [58]. In addition, large cities with intelligent pollution monitoring platforms can improve the city’s monitoring of enterprise emissions and enterprises’ monitoring of their own emissions, in addition to the ability to obtain pollution information through real-time monitoring, thus alleviating the haze emissions [59]. In small and medium-sized cities, the implementation of environmental regulations and technical effects are relatively weak, and the environmental monitoring mechanism is not perfect. Therefore, the haze reduction effect is not obvious.

## 6. Further Analysis

The above empirical results use the general DID model to identify the causal relationship between LCCP and haze pollution without considering the spatial effect of LCCP on haze pollution. In this section, we present the results of spatial econometric methods to further investigate whether the local LCCP has a spatial spillover effect on haze pollution in adjacent cities.

### 6.1. Global Spatial Correlation Test

In this study, we selected the adjacency matrix and used the global Moran I to test the spatial correlation of haze pollution [23]. The results in Table 10 show that the Moran index of haze pollution fluctuated between 0.179 and 0.221 from 2005 to 2018, and all values passed the significance test at the significance level of 1%, which indicates that there is a significant positive spatial correlation in haze pollution.

### 6.2. Spatial Econometric Model Test and Result Analysis

Based on the verified spatial correlation, in this study, we initially set the model as a two-way fixed spatial Durbin model (SDM) and carried out a correlation test. The LM and robust LM tests rejected the original hypothesis of the non-spatial effect model. In addition, the LR test rejected the original hypothesis of joint non-significance of spatial and temporal fixed effects, the Hausman test rejected the random effect model, and the Wald test rejected the original hypothesis that the spatial Durbin model was transformed into a spatial lag and spatial error model. According to the test results in Table 11, the dynamic spatial Durbin model with spatiotemporal double fixed effects was selected.

In Table 12, Column (1) shows the overall regression results. It can be seen that the spatial autoregressive coefficient is significantly positive at the level of 1%, indicating that there is a significant positive spatial dependence in haze pollution, which further verifies the rationality of introducing spatial factors into the analysis. The estimated coefficient of LCCP is significantly negative, implying that LCCP inhibits haze pollution. In addition, the regression coefficient of the explanatory variable in the spatial econometric model cannot directly reflect the impact on the explained variable [60]. This needs to be decomposed into the direct effect, indirect effect, and total effect by calculating the partial differentials. The estimation results are shown in columns (2)–(4). The direct effect reflects that the LCCP has a significant inhibitory effect on haze pollution, which is consistent with the benchmark regression conclusion. The indirect effect, i.e., the spillover effect, shows that LCCP can inhibit haze pollution in adjacent cities [19]. This implies that LCCP has a demonstration and warning effect, which forces neighboring cities to strengthen pollution prevention and controls to curb haze pollution, such as eliminating backward production capacity, upgrading the industrial structure, and driving technological innovation [61].

## 7. Conclusions and Policy Recommendations

Based on the panel data of 271 prefecture-level cities in China from 2005 to 2018, in this study we employed the DID and mediatory effect models to empirically test the impact and mechanism of LCCP on urban haze pollution. In addition, heterogeneity analysis was undertaken to further explore the differential impacts from urban resource dependence, geographic location, and scale. Finally, the spatial spillover effect of LCCP on haze pollution was further explored. The main conclusions are as follows: (1) LCCP has significantly inhibited urban haze pollution, reducing the average annual concentration of PM2.5 by 14.29%. After a series of robustness tests, this conclusion was found to be valid. (2) The mechanism analysis shows that LCCP reduces haze pollution through technological innovation, industrial structure, and energy consumption. Among these, technological innovation has the strongest effect, followed by industrial structure and energy consumption. (3) Heterogeneity analysis implies that LCCP has significantly reduced the haze pollution of non-resource dependent cities, Eastern cities, and large cities, but has had little impact on resource-dependent cities, Central and Western regions, and small and medium-sized cities. (4) Further analysis shows that LCCP has a spatial spillover effect; that is, it can inhibit the haze pollution of adjacent cities through demonstration and warning effects.

Accordingly, the following policy implications are proposed:

(1) The scope of low-carbon pilot cities should be further expanded to promote the construction of low-carbon cities. The government should summarize the successful experience of low-carbon city pilot construction, encourage the non-pilot cities to learn from the pilot cities, and constantly explore the new path of urban low-carbon development. (2) The government should fully utilize the role of LCCP in haze control. Specifically, the government should strengthen the assessment of environmental protection and energy consumption in the construction of low-carbon pilot cities, improve the intensity of environmental regulation, and actively implement fiscal and tax policies and relevant incentive policies that are conducive to enterprise scientific and technological innovation, industrial structure optimization, and upgrading of the energy structure, so as to improve the governance efficiency of haze pollution. (3) The government should further promote the construction of low-carbon pilot cities in an orderly manner, and give priority to the establishment of low-carbon cities in the Eastern region, and non-resource-based and large-scale cities. The cities in the Central and Western regions, in addition to small-scale and resource-based cities, should learn and absorb the haze controlling experience from the pilot cities in the Eastern region, and non-resource-based and large-scale cities. In addition, the government should give more policy preference and support to cities in the Central and Western regions and small-sized and resource-based cities, improve their pilot policy assessment system and environmental monitoring mechanism, and finally fully utilize the haze reduction effect of LCCP. (4) The cross-regional cooperation among pilot cities should be promoted for haze prevention and control. The government should abandon the traditional territorial management mode of “each doing its own thing” and “beggar thy neighbor”. Alternatively, it should create a “decentralized” and “networked” cross-regional joint prevention and control mode, and build a normalized regional feedback mechanism and information sharing platform, so as to strengthen cross-regional collaborative governance of haze prevention and control [53]. Additionally, pilot cities should fully utilize their radiation and driving role, based on their experience of innovative technology and environmental governance, for adjacent non-pilot cities, maximizing the spillover effect of LCCP on haze reduction.

Although this study explored the impact of LCCP on urban haze pollution and its impact mechanism, the exploration was not sufficient in terms of its depth and breadth due to the short time of implementation and the limitation in data. In the future, the long-term dynamic impact of the policy should be examined to further explore additional means to alleviate haze pollution.

## Figures and Tables

**Figure 1 ijerph-18-11287-f001:**
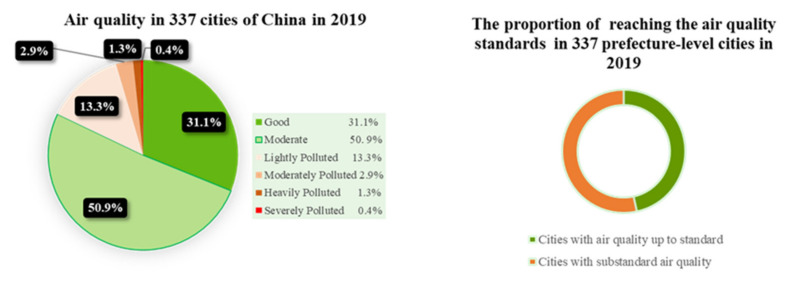
Proportion of air quality days at all levels and quality standards in 337 cities in 2019.

**Figure 2 ijerph-18-11287-f002:**
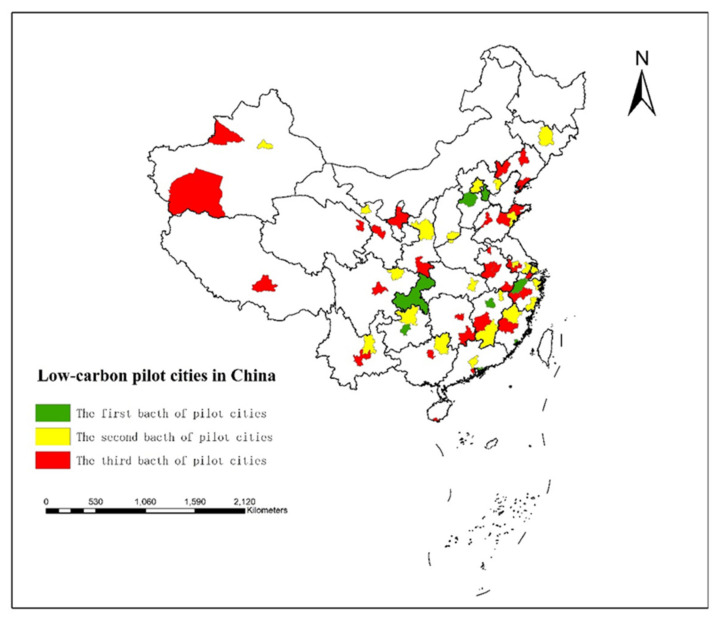
Spatial distribution of the three batches of low-carbon pilot cities.

**Figure 3 ijerph-18-11287-f003:**
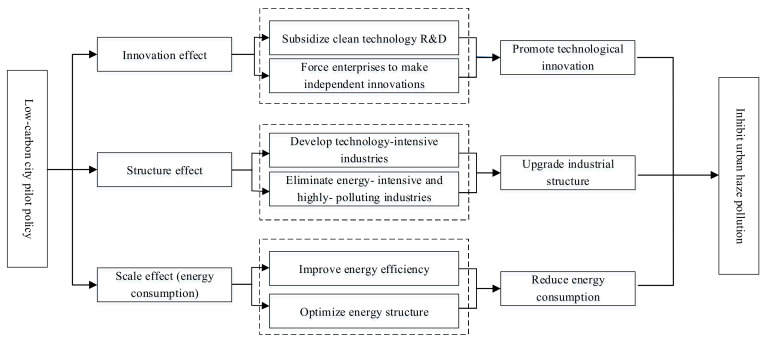
Mechanism diagram of the impact of LCCP on haze pollution.

**Figure 4 ijerph-18-11287-f004:**
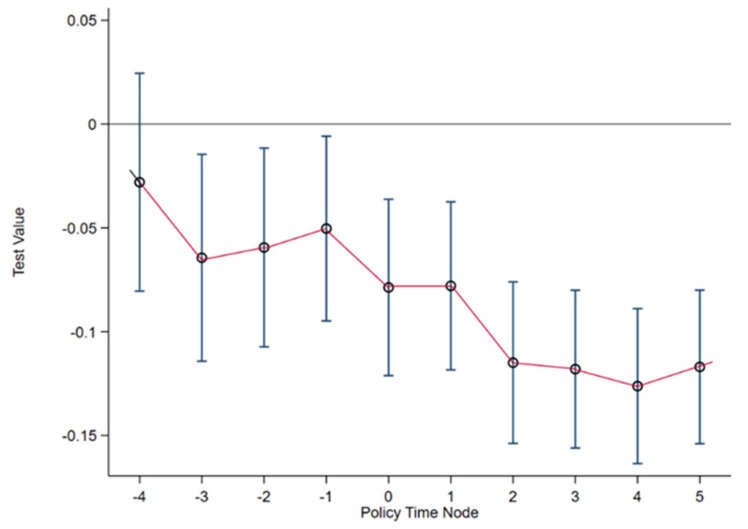
Parallel trend test chart.

**Figure 5 ijerph-18-11287-f005:**
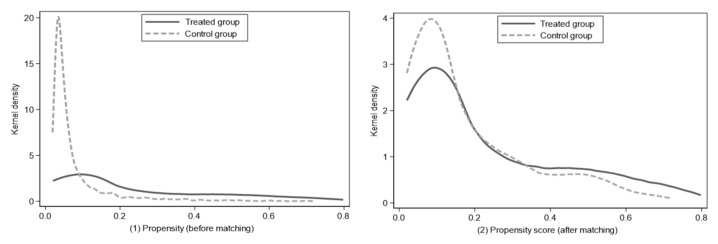
Kernel density function matching diagram.

**Table 1 ijerph-18-11287-t001:** Variable description statistics.

Variable	Obs	Mean	Std. Dev.	Min	Max
lnPM_2.5_	3780	3.5030	0.5038	1.5425	4.5093
LCCP	3780	0.0780	0.2682	0	1
lnRGDP	3780	10.3966	0.7572	8.1098	12.9596
lnHUM	3779	−0.1486	1.3729	−36.8414	2.7044
lnFDI	3737	−0.1466	1.3627	−10.2046	4.3428
lnFIN	3780	−0.3187	0.5026	−2.011	2.8402
lnINFOR	3771	−2.3272	0.9628	−6.309	1.0286
lnPATENT	3763	6.5083	1.8727	0	11.9045
lnIND	3780	3.5996	0.2468	2.1494	4.4466
lnPELE	3780	7.0711	1.1950	3.5653	11.56

**Table 2 ijerph-18-11287-t002:** Benchmark regression results.

Variable	(1)	(2)	(3)	(4)	(5)	(6)
	lnPM_2.5_	lnPM_2.5_	lnPM_2.5_	lnPM_2.5_	lnPM_2.5_	lnPM_2.5_
LCCP	−0.0740 **	−0.0670 **	−0.0883 ***	−0.1040 ***	−0.0763 ***	−0.1429 ***
	(−2.42)	(−2.09)	(−2.77)	(−3.37)	(−2.66)	(−1.88)
lnRGDP		−0.0082	−0.0511 ***	−0.0743 ***	−0.0866 ***	−0.1558 ***
		(−0.72)	(−4.12)	(−6.16)	(−6.68)	(−9.38)
lnHUM			−0.0546 ***	−0.0297 ***	−0.0520 ***	−0.0611 ***
			(−8.19)	(−4.51)	(−8.44)	(−9.70)
lnFDI				0.1077 ***	0.1113 ***	0.1143 ***
				(17.74)	(19.74)	(20.29)
lnFIN					−0.4540 ***	−0.4453 ***
					(−24.60)	(−24.15)
lnINFOR						−0.0843 ***
						(−6.59)
Cons	3.5088 ***	3.5936 ***	4.0497 ***	4.2742 ***	2.4586 ***	1.5450 ***
	(41.39)	(30.60)	(31.37)	(34.09)	(17.85)	(7.94)
Time-fixed	Yes	Yes	Yes	Yes	Yes	Yes
City-fixed	Yes	Yes	Yes	Yes	Yes	Yes
N	3780	3780	3779	3736	3736	3727
R^2^	0.002	0.002	0.019	0.095	0.221	0.231
F	5.870	3.197	24.54	97.81	211.9	186.2

Note: ** and *** indicate significance at the 5% and 1% levels respectively. Z-values are in parentheses.

**Table 3 ijerph-18-11287-t003:** Applicability test of the PSM-DID method.

	Mean		%Bias	T-Value	*p*-Value
Variable	Treated	Control			
lnRGDP	11.131	11.161	−4.6	−0.67	0.504
lnHUM	0.7967	0.8024	−0.5	−0.07	0.947
lnFDI	0.7356	0.7472	−0.9	−0.12	0.906
lnFIN	0.0209	0.062	−7.9	−0.9	0.366
lnINFOR	−1.3386	−1.318	−2.5	−0.34	0.731

**Table 4 ijerph-18-11287-t004:** Estimation results by PSM-DID method.

Variables	(1)	(2)	(3)	(4)	(5)	(6)
	lnPM2.5	lnPM2.5	lnPM2.5	lnPM2.5	lnPM2.5	lnPM2.5
LCCP	−0.0877 ***	−0.1947 ***	−0.1933 ***	−0.2057 ***	−0.1997 ***	−0.1529 ***
	(0.0319)	(0.0314)	(0.0314)	(0.0307)	(0.0288)	(0.0282)
lnRGDP		−0.2527 ***	−0.2550 ***	−0.2383 ***	−0.0505 **	0.0169
		(0.0201)	(0.0201)	(0.0197)	(0.0209)	(0.0211)
lnHUM			−0.0292 ***	−0.0505 ***	0.0104	0.0146
			(0.0107)	(0.0105)	(0.0104)	(0.0101)
lnFDI				0.0948 ***	0.1078 ***	0.1101 ***
				(0.0070)	(0.0066)	(0.0065)
lnFIN					−0.4057 ***	−0.3692 ***
					(0.0211)	(0.0208)
lnINFOR						−0.2050 ***
						(0.0165)
Cons	3.5199 ***	5.6130 ***	5.5900 ***	5.4297 ***	3.4117 ***	2.1826 ***
	(0.0102)	(0.2000)	(0.2000)	(0.1955)	(0.2114)	(0.2286)
N	2780	2779	2778	2735	2735	2726
R^2^	0.0030	0.1340	0.1360	0.1890	0.2860	0.3250

Note: ** and *** indicate significance at the 5% and 1% levels respectively. Standard errors are in parentheses.

**Table 5 ijerph-18-11287-t005:** Transmission mechanism test.

	(1)	(2)	(3)	(4)	(5)	(6)	(7)
Intermediary Effect		Technological Innovation		Industrial Structure		Energy Consumption	
Explained Variable	ln*PM2.5*	ln*PATENT*	ln*PM2.5*	ln*IND*	ln*PM2.5*	ln*PELE*	ln*PM2.5*
LCCP	−0.1429 ***	0.3534 ***	−0.1634 ***	0.5445 ***	−0.124 ***	−0.1967 ***	−0.1371 ***
	(0.0274)	(0.0252)	(0.0281)	(0.136)	(0.0271)	(0.05)	(0.0275)
M			−0.058 ***		−0.3474 ***		0.0294 ***
			(0.0178)		(0.0326)		(0.009)
Control	Yes	Yes	Yes	Yes	Yes	Yes	Yes
Cons	2.3001 ***	−0.4538 ***	2.3270 ***	4.9035 ***	4.0044 ***	207,813 ***	−5.1346 ***
	(0.1873)	(0.172)	(0.054)	(0.093)	(0.2441)	(0.3414)	(0.225)
Sobel		0.0205		−0.0189		−0.0058	
		(Z = 3.165, *p* = 0.0015)		(Z = −3.749, *p* = 0.000)		(Z = −2.517, *p* = 0.0118)	
Bootstrap test		0.1634		0.124		0.1371	
(direct effect)		(Z = −5.8112, *p* = 0.000)		(Z = −0.0271, *p* = 0.000)		(Z = −4.9945, *p* = 0.000)	
Bootstrap test		0.0205		0.0189		0.0059	
(indirect effect)		(Z = 3.165, *p* = 0.0015)		(Z = −3.749, *p* = 0.000)		(Z = −2.517, *p* = 0.0118)	
Indirect effect proportion		14.36%		13.23%		4.06%	
R^2^	0.3143	0.8872	0.3163	0.2942	0.3347	0.5967	0.3163

Note: *** indicates significance at the 1% level. Standard errors are in parentheses.

**Table 6 ijerph-18-11287-t006:** Estimation results of the robustness test.

Variable	(1)	(2)	(3)	(4)
	Placebo Test	SO2 as Explained Variable	Excluding 31 Key Cities	All Variables Lag For One Period
LCCP	−0.047	−0.2449 ***	−0.1321 ***	−0.1145 **
	(0.1612)	(0.0900)	(0.0439)	(0.0497)
Control	Yes	Yes	Yes	Yes
Cons	−4.545 ***	4.8123 ***	3.0562 ***	3.0519 ***
	(0.2231)	(0.8963)	(0.4263)	(0.4829)
Time-fixed	Yes	Yes	Yes	Yes
City-fixed	Yes	Yes	Yes	Yes
R^2^	0.3599	0.1279	0.3185	0.3371

Note: ** and *** indicate significance at the 5% and 1% levels respectively. Standard errors are in parentheses.

**Table 7 ijerph-18-11287-t007:** Estimation results of analysis of urban resource dependence heterogeneity.

Variable	Non-Resource-Based		Resource-Based	
	(1)	(2)	(3)	(4)
LCCP	−0.2049 ***	−0.2119 ***	−0.0542	0.00059
	(0.06200)	(0.0501)	(0.3395)	(0.03438)
Control	No	Yes	No	Yes
Cons	3.4325 ***	1.1012 ***	3.5622 ***	1.5578 ***
	(0.0139)	(0.2936)	(0.0105)	(0.2633)
Time-fixed	Yes	Yes	Yes	Yes
City-fixed	Yes	Yes	Yes	Yes
N	1502	1502	2268	2225
R^2^	0.0072	0.4019	0.0011	0.1440

Note: *** indicates significance at the 1% level. Standard errors are in parentheses.

**Table 8 ijerph-18-11287-t008:** Estimation results of analysis of urban regional heterogeneity.

Variable	Urban Regional Heterogeneity			
	Eastern Region		Central and Western Region	
	(1)	(2)	(3)	(4)
LCCP	−0.1557 ***	−0.1643 ***	−0.2413 ***	−0.0261
	(0.0452)	(0.0416)	(0.0509)	(0.0446)
Control	No	Yes	No	Yes
Cons	3.4552 ***	1.2763 ***	3.5111 ***	2.2169 ***
	(0.0322)	(0.3352)	(0.014)	(0.3199)
Time-fixed	Yes	Yes	Yes	Yes
City-fixed	Yes	Yes	Yes	Yes
N	1330	1297	1302	1286
R^2^	0.0365	0.2077	0.017	0.3629

Note: *** indicates significance at the 1% level. Standard errors are in parentheses.

**Table 9 ijerph-18-11287-t009:** Estimation results of analysis of urban scale heterogeneity.

Variable	Urban Scale Heterogeneity		
	(1) Large-Sized	(2) Middle-Sized	(3) Small-Sized
LCCP	−0.1148 ***	−0.3041	−0.0753
	(0.0274)	(0.2888)	(0.0941)
Control	Yes	Yes	Yes
Cons	2.0387 ***	5.7142 ***	2.6142 ***
	(0.187)	(1.6989)	(1.0726)
Time-fixed	Yes	Yes	Yes
City-fixed	Yes	Yes	Yes
N	3561	116	49
R^2^	0.2811	0.2457	0.4764

Note: *** indicates significance at the 1% level. Standard errors are in parentheses.

**Table 10 ijerph-18-11287-t010:** Moran’s I test results of haze pollution in Chinese cities (2005–2018).

Year	2005	2006	2007	2008	2009	2010	2011
Moran’s I	0.189 ***	0.179 ***	0.201 ***	0.190 ***	0.182 ***	0.186 ***	0.187 ***
Z-value	(25.739)	(24.467)	(27.322)	(25.837)	(24.853)	(25.415)	(25.494)
Year	2012	2013	2014	2015	2016	2017	2018
Moran’s I	0.181 ***	0.192 ***	0.184 ***	0.221 ***	0.204 ***	0.211 ***	0.218 ***
Z-value	(24.705)	(26.185)	(25.036)	(30.082)	(27.767)	(28.785)	(29.591)

Note: *** indicates significance at the 1% level. Figures in () are Z-values.

**Table 11 ijerph-18-11287-t011:** Spatial econometric model test.

Index	Value	*p*-Value	Index	Value	*p*-Value
LM-lag	5711.605	0.000	LM-error	7663.016	0.000
Robust LM-lag	19.549	0.000	Robust LM-error	1970.960	0.000
LR-lag	250.19	0.000	LR-error	264.85	0.000
WALD-SAR	258.34	0.000	WALD-SEM	273.89	0.000
Hausman	4.92	0.002			

**Table 12 ijerph-18-11287-t012:** Spatial econometric estimation and decomposition results.

Variable	Haze Pollution			
	（1）	（2）	（3）	（4）
	Main	LR-Direct	LR-Indirect	LR-Total
LCCP	−0.2217 **	−0.1528 **	−0.0871 *	−0.2345 **
	(−3.14)	(−1.71)	(−1.81)	(−3.52)
*ρ*	0.9861 ***			
	(266.09)			
sigma2_e	0.0646 ***			
	(4.315)			
Control	Yes	Yes	Yes	Yes
Time-fixed	Yes	Yes	Yes	Yes
City-fixed	Yes	Yes	Yes	Yes
R2	0.041	0.041	0.041	0.041

Note: *, ** and *** are significant at 10%, 5% and 1% levels respectively. Z-values are in parentheses.

## Data Availability

Not applicable. No new data were created or analyzed in this study.

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
