# Peer review of "Does Low-Carbon City Pilot Policy Alleviate Urban Haze Pollution? Empirical Evidence from a Quasi-Natural Experiment in China"

_ijerph, 2021, doi:10.3390/ijerph182111287_

Round 1

Reviewer 1 Report

The topic is of interest. The “story telling” is convincing. The econometric methodology is technically implemented well. Nonetheless, I think that the paper needs significant improvement from both the economic and econometric point of view before being published.

1、The literature contribution should be explained in the Introduction, I do not know that the core questions of this study?

2、I believe that two hypotheses should be one by one clarified, especially, the second  hypothesis should be clearly explained by adding related content.

3、In this Literature review, I find that there are so many Chinese literatures, the adequate English papers should be presented in this section. More seriously, the literature review is so short.

4、The policy background should be explained in Introduction.

5、There are a dearth of the evidence regarding the selection of control variables, why these variables were selected?

6、I believe that there are spatial correlation in urban haze pollution. Therefore, the spatial econometric model should be adopted in this study.

7、More seriously, there is a lack of discussion regarding the empirical results, I do not know that there are literature contributions?

8、I think that this research should be polish by professional institution. The language is too hard to read and understand.

9、The scale unit in Figure 2 should be kilometers instead of miles.

Overall, I think that the study should be major revision before publishing. 

Author Response

Dear  Reviewer,

Thank you very much for carefully reading the manuscript. We are very grateful to your helpful and valuable recommendations for further improvements of the paper. We have revised the manuscript according to the reviewers’ comments and suggestions. All revisions to the manuscript have been marked up using the “Track Changes” function such that any changes can be easily viewed by the editors and reviewers and the answers to the reviewers’ comments have been marked dark blue in this response letter. 

Reviewer 2 Report

The literature review is incomplete. Please refer to the larger number of thematically related scientific publications published in recent years. 

Please adapt the manuscript to the journal's requirements, e.g. the year of publication of an article from a journal in References should be bold, the year of publication (References) should not be placed in parentheses and should be elsewhere, cited articles should be placed as reference numbers from References placed in square brackets in the manuscript text. the manuscript appears to have been formatted for a different journal.

Some parts of the Conclusions chapter seem too obvious: "The government should vigorously develop technology-intensive and environment-friendly industries to promote the transformation of industrial structure in the direction of being green and efficient. In addition, the government should encourage investment in energy technology innovation, pay more attention to cultivating energy scientific and technological talents, support the development of energy-saving and efficient industries, creating a good policy environment for the development and utilization of energy-saving technologies to reduce energy consumption." Please consider correcting them. In the Conclusions chapter, please emphasize the importance of the analyzes carried out in the article. 

Author Response

(The authors gave the same response as above.)

Round 2

Reviewer 1 Report

Following these comments, the author have revisied the manuscript. I believe that the article can be accepted now. However, the distinguishability of all pictures should be improved before publishing.

Reviewer 2 Report

The revised manuscript may be published in present form.